# UNDERSTANDING THE SUCCESS OF KNOWLEDGE DISTILLATION – A DATA AUGMENTATION PERSPECTIVE

## ABSTRACT

Knowledge distillation (KD) is a general neural network training approach that uses a teacher model to guide a student model. Many works have explored the rationale for its success. However, its interplay with data augmentation (DA) has not been well understood so far. In this paper, we are motivated by an interesting observation in classification: KD loss can take more advantage of a DA method than cross-entropy loss *simply by training for more iterations*. We present a generic framework to explain this interplay between KD and DA. Inspired by it, we enhance KD via stronger data augmentation schemes named TLmixup and TLCutMix. Furthermore, an even stronger and efficient DA approach is developed specifically for KD based on the idea of active learning. The findings and merits of our method are validated with extensive experiments on CIFAR-100, Tiny ImageNet, and ImageNet datasets. We achieve new state-of-the-art accuracy by using the original KD loss armed with stronger augmentation schemes, compared to existing state-of-the-art methods that employ more advanced distillation losses. We also show that, by combining our approaches with the advanced distillation losses, we can advance the state-of-the-art even further. In addition to very promising performance, this paper importantly sheds light on explaining the success of knowledge distillation. The interaction of KD and DA methods we have discovered can inspire more powerful KD algorithms.

## 1 INTRODUCTION

Deep neural networks (DNNs) are the best performing machine learning method in many fields of interest (LeCun et al., 2015; Schmidhuber, 2015). How to effectively train a deep network for classification has been a central topic for decades. In the past several years, efforts have mainly focused on better architecture design (*e.g.*, batch normalization (Ioffe & Szegedy, 2015), residual blocks (He et al., 2016), dense connections (Huang et al., 2017)) and better loss functions (*e.g.*, label smoothing (Szegedy et al., 2016; Müller et al., 2019), contrastive loss (Hinton, 2002), large-margin softmax (Liu et al., 2016)) than the standard cross-entropy (CE) loss. Knowledge distillation (KD) (Hinton et al., 2014) is a training framework that falls in the second group. In KD, a stronger network – called teacher – is introduced to guide the learning of the original network – called student – by minimizing the discrepancy between the representations of the two networks,

$$\mathcal{L}_{KD} = (1 - \alpha)\mathcal{L}_{CE}(y, p^{(s)}) + \alpha\tau^2 \mathcal{D}_{KL}(p^{(t)}/\tau, p^{(s)}/\tau), \qquad (1)$$

where $\mathcal{D}_{KL}$ represents KL divergence (Kullback, 1997); $\alpha \in (0, 1)$ is a factor to balance the two loss terms; $\mathcal{L}_{CE}$ denotes the cross-entropy loss; $y$ is the one-hot label and $p^{(t)}, p^{(s)}$ stands for the teacher's output and student's output, respectively (which are probability distributions over the classes); $\tau$ is a temperature constant (Hinton et al., 2014) to smooth predicted probabilities. KD allows us to train smaller, more efficient neural networks without compromising on accuracy, which facilitates deploying deep learning in resource constrained environments (*e.g.*, on mobile devices). The effectiveness of KD has been seen in many tasks (Chen et al., 2017; Wang et al., 2020; Jiao et al., 2019; Wang & Yoon, 2021). Meanwhile, many works have investigated the reason behind its success, such as class similarity structure (Hinton et al., 2014) and regularization (Yuan et al., 2020). However, few works have paid attention to its interplay with the input image data augmentation (DA), a technique to obtain more data through various transformations (Shorten & Khoshgoftaar, 2019). In this paper, we will show that data augmentation is also an important dimension to explain

the success of KD. Moreover, our findings show we can achieve much better performance simply using the original KD loss equipped with a stronger data augmentation scheme.

Our proposed algorithms are inspired from interesting observations shown in Fig. 1, where we plot the student test error curves when the model is trained for different numbers of epochs using KD loss vs. CE loss[1]. Three data augmentation scenarios are examined: not using DA at all (Without DA), only using the horizontal flip (Flip), and using both the horizontal flip and random crop (Flip+Crop). We have the following observations. **(1)** Within each plot, KD loss delivers lower test error than CE loss. **(2)** When DA is used (comparing the middle or right plot to the left), both CE and KD curves are improved. **(3)** When DA is used (comparing the middle or right plot to the left), the optimal number of training epochs is postponed and the postponement is greater for KD than CE (the optimal number of epochs is postponed from 180 to 480 for KD versu from 60 to 120 for CE). **(4)** When a *stronger* DA is employed (comparing the right plot to the middle), the optimal number of epochs is *further postponed* with even lower test error. The first two obser-

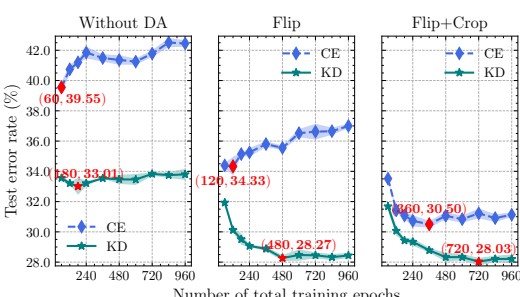

Figure 1: Test error rate of ResNet20 on CIFAR-100 when trained for different numbers of epochs (the teacher is ResNet56 for KD). Each result is obtained by averaging 3 random runs (shaded area indicates the std) "Flip" refers to horizontal flip; "Crop" refers to random crop. Both are standard data augmentation schemes in classification. The optimal number of training epochs and its test error are highlighted in red.

vations are well-recognized by existing works (they simply reiterate the effectiveness of KD and DA, respectively), while the last two observations are new discoveries of this work, concerning the *interaction* between KD and DA. In other words, KD and DA, as two common techniques to improve the performance of DNNs, are actually ***not independent***. This paper explains the interplay between KD and DA and leverages it for stronger KD performance using only the standard KD loss.

Specifically, we explain why KD is able to exploit DA more than the CE loss. Owing to the random transformations in data augmentation, the input data are *not fixed* over different epochs. *Different views* of each image are presented over the training process. When KD loss is used, the teacher maps these different views to different targets. As illustrated in Fig. 2(a), these targets with different probability structures can reveal more information of the data, thus helping the student more. In contrast, when CE loss is adopted, the target is fixed regardless of the different views of the input. The extra information is thus lost. This observation inspired us to develop two stronger data augmentation techniques (TLmixup, TLCutmix) that are tailored for KD. We further tap into the idea of active learning to make TLCutmix even better, that is, our TLCutmix+pick method.

In summary, these are the main contributions of this work:

- We make novel observations of the interaction between KD and DA. We explain how DA methods are more suited to be exploited by KD which uses teacher outputs as labels instead of the ground-truth one-hot labels used by CE loss.

- Inspired by the above, we propose to enhance the original KD loss with stronger data augmentation schemes (by adapting mixup (Zhang et al., 2018) and CutMix (Yun et al., 2019) to KD). It is shown that these methods are more reasonably applied in the KD case than in the CE case.

- We further propose an even stronger data augmentation method specifically for knowledge distillation using the idea of active learning (Settles, 2011).

- We show empirically better results simply by using the original KD loss combined with the proposed DA scheme, compared to state-of-the-art KD methods, which adopt more advanced distillation losses.

---

[1]Note, data points on one curve in Fig. 1 are from *independent experiments of different total number of epochs*; the learning rate schedule is proportionally scaled based on the total number of epochs.

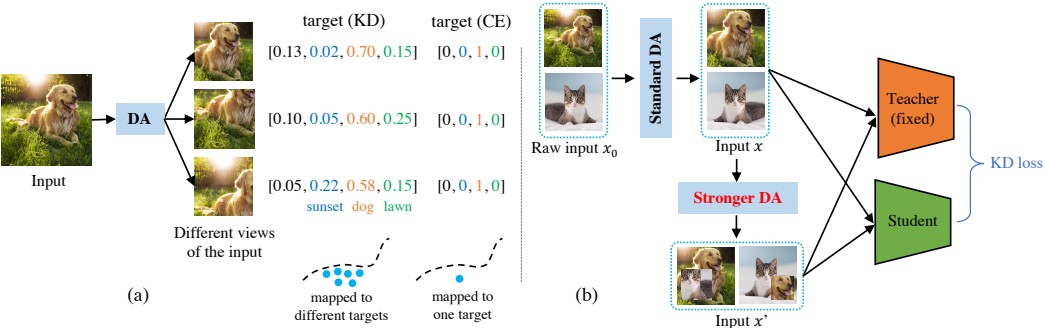

Figure 2: Interplay between knowledge distillation (KD) and data augmentation (DA). **(a)** Illustration of the difference of supervised target between the KD loss and cross-entropy (CE) loss. An input is transformed to different versions (called *views* in this paper) owing to data augmentation. KD loss can provide extra information to the student by mapping these views to different targets, while the CE loss cannot. **(b)** Illustration of KD with the proposed data augmentation framework. The standard DA consists of random crop and horizontal flip. The stronger DA refers to any data augmentation scheme more advanced than the standard one. In this paper, we propose three stronger DA schemes: TLmixup, TLCutMix, TLCutMix+pick (see Sec. 3.2 for details).

## 2 RELATED WORK

**Knowledge distillation:** The general idea of knowledge distillation is to guide the training of a student model through a stronger pretrained teacher model (or an ensemble of models). It was pioneered by Buciluǎ *et al.* (Buciluǎ et al., 2006) and later refined by Hinton *et al.* (Hinton et al., 2014), who coined the term. Since its debut, knowledge distillation has seen extensive application in vision and language tasks (Chen et al., 2017; Wang et al., 2020; Jiao et al., 2019; Wang & Yoon, 2021). Many variants have been proposed regarding the central question in knowledge distillation, that is, how to define the *knowledge* that is supposed to be transferred from the teacher to the student. Examples of such knowledge definitions include feature distance (Romero et al., 2015), feature map attention (Zagoruyko & Komodakis, 2017), feature distribution (Passalis & Tefas, 2018), activation boundary (Heo et al., 2019), inter-sample distance relationship (Park et al., 2019; Peng et al., 2019; Liu et al., 2019; Tung & Mori, 2019), and mutual information (Tian et al., 2020). Over the past several years, the progress has been made primarily at the *output end* (*i.e.*, through a better loss function). In contrast to previous works, our goal in this paper is to improve the KD performance at the *input end* with the help of data augmentation. We will show this path is as effective and also has much potential for future work.

**Data augmentation:** Deep neural networks are prone to overfitting, *i.e.*, building input-target connection using undesirable or irrelevant features (like noise) in the data. Data augmentation is a prevailing technique to curb overfitting (Shorten & Khoshgoftaar, 2019). In classification tasks, data augmentation aims to explicitly provide data with label-invariant transformations (such as random crop, horizontal flip, color jittering) in the training so that the model can learn representations robust to those nuisance factors. Recently, some advanced data augmentation methods were proposed, which not only transform the input, but also transform the target based on certain corresponding relations. For example, mixup (Zhang et al., 2018) linearly mixes two images with the labels mixed by the same linear interpolation; manifold mixup (Verma et al., 2019) is similar to mixup but conducts the mix operation in the feature level instead of pixel level; CutMix (Yun et al., 2019) pastes a patch cut from an image onto another image with the label decided by the area ratio of the two parts. Now that the input and target are transformed simultaneously, the key is to maintain a *semantic correspondence* between the new input and new target. Although these methods have been proven effective, one lingering concern is about the reasoning behind them. Specifically, it is easy to come up with examples where the semantic correspondence is poorly kept (see Fig. 5 for examples on CutMix). Unlike these methods, which focus on general classification using the cross-entropy loss, our work investigates the interplay between data augmentation and knowledge distillation loss and the proposed new data augmentation is specifically for knowledge distillation.

One recent work (Das et al., 2020) also conducts empirical study about the impact of data augmentation on knowledge distillation. However, their exploration is very different from ours: they first apply data augmentation (*e.g.*, mixup/CutMix) to the teacher then conduct the distillation step as usual (no extra data augmentation in this step); our investigation is the *exact opposite* to their setup: we train the teacher as usual (not applying mixup/CutMix), then in the distillation step we employ a more advanced data augmentation (*e.g.*, mixup/CutMix). Interestingly, they conclude that the teacher trained with mixup/CutMix *hurts* the student's generalization ability, while we consistently see a performance boost by using a stronger DA in the distillation step during student training.

## 3 PROPOSED METHOD

### 3.1 INTERPLAY OF KD, DA, AND TRAINING ITERATIONS

We first introduce a framework to explain the phenomenon that KD can exploit DA more simply by training for more iterations. Training for more iterations means presenting more examples to the network. Over iterative training, the presented examples are not exactly the same among different epochs because of the random transformations of data augmentation. Different versions of an image produced by data augmentation can be regarded as multi-views of that image (Wu et al., 2018; Tian et al., 2020). We term this kind of data difference as *input view diversity*. As depicted in Fig. 2(a), when the CE loss is employed, different views of an image are mapped to *a single point* in the target space (the hard label). In contrast, when the KD loss is used, different views of the data are mapped to *a group of points* in the target space through the teacher, which can reveal richer information around that class. By *richer*, we specifically mean two sources. First is the class structure information provided by the soft labels instead of the hard labels, *i.e.*, the well-known dark knowledge (Hinton et al., 2014). Second is the information provided by the different input views from data augmentation.

Concretely, in Fig. 2(a), although the three views of the input share the same main class "dog", the target probability vectors are different: compared with the first view, the second one has more lawn in it, thus the teacher has larger predicted probability in the "lawn" class; similarly, the third view has more sunshine, thus larger probability in the "sunset" class. These subtle changes in class-wise probabilities are beneficial for the student to learn. However, if the CE loss is used, all the three views are mapped to the same one-hot label, not putting the extra information to good use. More training iterations keep producing new data views to the student, making it afford more training epochs without overfitting. In contrast, for CE, as the target is fixed for different views, little new information is added in longer training. Thus, the student can only afford fewer extra epochs.

Denote the optimal number of training epochs as $N$, lowest test error as $E$. The synergistic interplay between KD and DA can be summarized in the following hypotheses,

$$N_{KD}^{(\text{DA})} - N_{KD}^{(\text{w/o DA})} > N_{CE}^{(\text{DA})} - N_{CE}^{(\text{w/o DA})},$$
$$N_{KD}^{(\text{DA+})} > N_{KD}^{(\text{DA})} \text{ and } E_{KD}^{(\text{DA+})} < E_{KD}^{(\text{DA})}, \tag{2}$$

where "DA" refers to a data augmentation method; "DA+" refers to another data augmentation method stronger than "DA"; "w/o DA" means not using any data augmentation. These inequalities are empirically verified in our experiments (Fig. 1, Fig. 3). The first inequality suggests the advantage of KD over CE: given the same DA scheme, KD loss can make *more use* of extra training iterations. The second suggests we can obtain better accuracy simply by training for more epochs using a *stronger* DA method. This leads us to investigate stronger DA methods for KD, as follows.

### 3.2 PROPOSED ALGORITHMS FOR IMPROVED KD

**(1) KD+TLmixup/TLCutMix**. We continue our exploration with two existing data augmentation techniques that are more advanced than the standard random crop and flip: mixup (Zhang et al., 2018) and CutMix (Yun et al., 2019). They are initially proposed in the CE case. Here we upgrade them for KD, resulting in *TLmixup* and *TLCutMix* (TL is short for teacher-labeled).

Specifically, let $x_0$ denote the raw data, $x$ denote the transformed data by the standard augmentation (random crop and flip). Illustrated in Fig. 2(b), we propose to add mixup/CutMix following $x$ to obtain $x'$. Unlike the common data augmentation where *only* the transformed input is fed into the

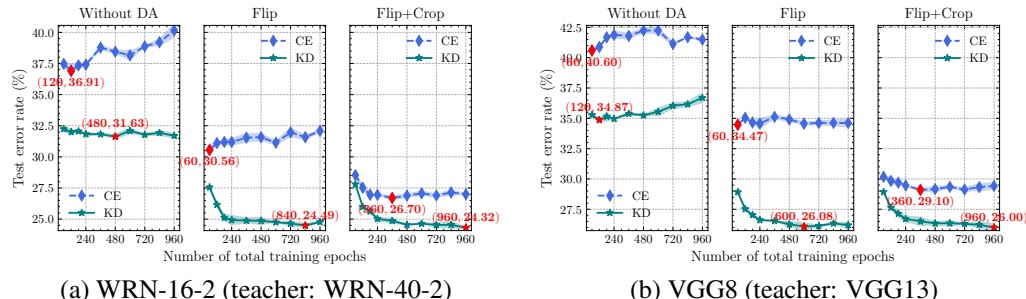

(a) WRN-16-2 (teacher: WRN-40-2)          (b) VGG8 (teacher: VGG13)

Figure 3: Test error rate of WRN-16-2 and VGG8 on CIFAR-100 when trained for different numbers of epochs, using KD or cross-entropy (CE) loss, with or without data augmentation (DA). Every error rate is averaged by 3 random runs (shaded area indicates the stddev). Consistent with Fig. 1, when DA is used, the optimal number of epochs is postponed and postponed more for KD than CE. When a stronger DA is used, the optimal number of epochs is postponed even more with smaller optimal test error.

network, we keep *both the input $x$ and $x'$* for the training (as such, the number of input examples during training is increased). The consideration of keeping both inputs is to maintain the information path for the original input $x$ so that we can easily see how the added information path of $x'$ leads to a difference.

For $x$, its loss is still the original KD loss, consisting of the cross-entropy loss and the KL divergence (Eq. 1). Of special note is that, for $x'$, its loss is *only* the KL divergence, *i.e.*, *we do not use the labels assigned by mixup or CutMix* because they can be misleading and do not perform well as we will show later (Tab. 1, Fig. 5). In fact, not using the hard label has another bonus. A dataset augmentation scheme which employs CE loss has to provide corresponding labels as supervisory information. In order to maintain the semantic correspondence, it cannot admit very extreme transformations for data augmentation. In contrast, in the mixup/CutMix+KD setting described above, the data augmentation scheme need not worry about the labels as they are assigned by the teacher. Therefore, it admits a *broader* set of transformations to expose the teacher's knowledge more completely.

Between TLmixup and TLCutMix, we will empirically show TLCutMix is more favorable (Tab. 1) (and both of them are significantly better than the standard augmentation, random crop and flip). Therefore, we choose TLCutMix as base to develop our next algorithm as follows.

**(2) KD+TLCutMix+pick**. Our next algorithm is an even stronger DA scheme tailored to KD, based on the idea of *active learning* (Settles, 2011). In active learning, the learner enjoys the freedom to query the data instances to be labeled for training by an oracle (*i.e.*, the teacher in our case) (Settles, 2011). Since the augmented data can vary in their quality, we can introduce certain criterion to pick the more valuable data for the student. We tap into the idea of hard examples (Micaelli & Storkey, 2019) to define the criterion. Specifically, we measure the hardness by the KL divergence between the teacher's output and the student's output ($p^{(t)}$ and $p^{(s)}$ are the teacher and student output probabilities over classes),

$$d = \mathcal{D}_{KL}(p^{(t)}/\tau, p^{(s)}/\tau). \tag{3}$$

We sort the augmented samples by their $d$'s in ascending order and pick a subset with the largest $d$'s. Notably, the criterion $d$ has exactly the same form that the student is supposed to *minimize* in Eq. (1); while here we pick samples to *maximize* it. This design makes an adaptive competition: when the student is updated, the criterion made of the KL divergence will also be updated. Each time, it makes sure the hardest samples are selected for the student.

Other common choices for the criterion include the teacher's entropy or the student's entropy (larger entropy implying more uncertainty meaning the sample is harder). They only take into account one-side information, either the teacher's or student's. Conceivably, they are not as good as the KL divergence criterion, which considers the information from both sides. This choice will be empirically justified in our experiments (Tab. 2).

Table 1: KD test accuracy comparison on CIFAR-100 when using different DA schemes. Each experiment is run 3 times and the mean and (standard deviation) are reported. Default DA: random crop and flip. Note (1) KD+CutMix is much worse than KD alone on Res56/Res20; (2) KD+TLCutMix consistently outperforms KD+CutMix on all the 3 pairs.

| Teacher
Student | WRN-40-2
WRN-16-2 | ResNet56
ResNet20 | VGG13
VGG8 |
|---|---|---|---|
| Teacher Acc. | 75.61 | 72.34 | 74.64 |
| Student Acc. | 73.26 | 69.06 | 70.36 |
| KD (default DA) | 74.92 (0.28) | 70.66 (0.24) | 72.98 (0.19) |
| KD+TLmixup | 75.33 (0.07) | **71.00** (0.16) | 73.79 (0.18) |
| KD+TLCutMix | **75.34** (0.19) | 70.77 (0.17) | **74.16** (0.18) |
| KD+CutMix | 75.25 (0.13) | 69.76 (0.19) | 73.75 (0.16) |

Table 2: KD test accuracy comparison on CIFAR-100 when using different data picking schemes for the proposed "KD+TLCutMix+pick" method. "T/S" refers to teacher/student, "ent." is short for entropy, and "kld" stands for KL divergence. The mean and (std) of 3 runs are reported for each entry.

| Teacher
Student | WRN-40-2
WRN-16-2 | ResNet56
ResNet20 | VGG13
VGG8 |
|---|---|---|---|
| Teacher Acc. | 75.61 | 72.34 | 74.64 |
| Student Acc. | 73.26 | 69.06 | 70.36 |
| TLCutMix | 75.34 (0.19) | 70.77 (0.17) | 74.16 (0.18) |
| +Pick (T ent.) | 75.46 (0.07) | 70.88 (0.10) | 74.16 (0.18) |
| +Pick (S ent.) | 75.52 (0.06) | 70.84 (0.12) | 74.16 (0.48) |
| +Pick (T/S kld) | **75.59** (0.22) | **70.99** (0.20) | **74.43** (0.20) |

## 4 EXPERIMENTAL RESULTS

**Datasets and networks**. We evaluate our method on the CIFAR-100 (Krizhevsky, 2009), Tiny ImageNet[2], and ImageNet (Deng et al., 2009) object recognition datasets. CIFAR-100 has 100 object classes ($32\times32$ RGB images). Each class has 500 images for training and 100 images for testing. ImageNet is now the standard large-scale benchmark dataset in image classification, which has 1000 classes ($224\times224$ RGB images), over 1.2 million images in total. Tiny ImageNet is a small version of ImageNet with 200 classes ($64\times64$ RGB images). Each class has 500 images for training, 50 for validation and 50 for testing. To thoroughly evaluate our methods, we benchmark them on various standard network architectures: VGG (Simonyan & Zisserman, 2015), ResNet (He et al., 2016), WRN (Wide-ResNet) (Zagoruyko & Komodakis, 2016), MobileNetV2 (Sandler et al., 2018), ShuffleNetV2 (Ma et al., 2018). *Our code and trained models will be made publicly available*.

**Evaluated methods**. In addition to the standard cross-entropy training and the original KD method (Hinton et al., 2014), we also compare with the state-of-the-art distillation approach Contrastive Representation Distillation (CRD) (Tian et al., 2020). It is important to note that our method focuses on improving KD by using better *inputs*, while CRD improves KD at the *output* end (*i.e.*, a better loss function). Therefore, they are orthogonal and we will show they can be combined together to deliver even better results.

**Hyperparameter settings**. The temperature $\tau$ of knowledge distillation is set to 4. Loss weight $\alpha = 0.9$ (Eq. equation 1). (1) For CIFAR-100 and Tiny ImageNet, training batch size is 64; the original number of total training epochs is 240, with learning rate decayed at epoch 150, 180, and 210 by multiplier 1/10. The initial lr is 0.05. (2) For ImageNet, training batch size is 256; the original number of training epochs is 100, with learning rate decayed at epoch 30, 60, 90. The initial learning rate is 0.1. All these settings are *the same* as CRD (Tian et al., 2020) for fair comparison with it. Note, in our experiments we will show the results of more training iterations. If the number of total epochs is scaled by a factor $k$, the epochs after which learning rate is decayed is also be scaled by $k$. For example, if we train a network for CIFAR-100 for 480 epochs ($k = 2$) in total, the epochs after which the learning rate is decayed will be 300, 360, and 420.

We use PyTorch (Paszke et al., 2019) to conduct all our experiments. For CIFAR-100, we adopt the pretrained teacher models from CRD (https://github.com/HobbitLong/RepDistiller) for fair comparison with it. For Tiny ImageNet, we train our own teacher models. For ImageNet, we adopt the standard torchvision models as teachers.

### 4.1 CIFAR-100

**Effect of more training iterations**. In Sec. 1, we presented Fig. 1 to show the advantage of KD loss over CE loss in exploiting extra epochs. Here we show more results in Fig. 3 on different network architectures to confirm the finding is general. In line with the ResNet case (Fig. 1), extra training

---

[2]https://tiny-imagenet.herokuapp.com/

Table 3: Student test accuracy comparison on CIFAR-100. Each result is obtained by 3 random runs, mean (std) accuracy reported. The best results are in **bold** and second best underlined. The subscript 960 means the total number of training epochs (default: 240).

| Teacher
Student | WRN-40-2
WRN-16-2 | ResNet56
ResNet20 | ResNet32x4
ResNet8x4 | VGG13
VGG8 | VGG13
MobileNetV2 | ResNet50
VGG8 | ResNet32x4
ShuffleNetV2 |
|---|---|---|---|---|---|---|---|
| Teacher Acc.
Student Acc. | 75.61
73.26 | 72.34
69.06 | 79.42
72.50 | 74.64
70.36 | 74.64
64.60 | 79.34
70.36 | 79.42
71.82 |
| KD (Hinton et al., 2014) | 74.92 (0.28) | 70.66 (0.24) | 73.33 (0.25) | 72.98 (0.19) | 67.37 (0.32) | 73.81 (0.13) | 74.45 (0.27) |
| KD$_{960}$ (Hinton et al., 2014) | 75.68 (0.12) | **71.79** (0.29) | 73.14 (0.06) | 74.00 (0.34) | 68.77 (0.05) | 74.04 (0.25) | 74.64 (0.30) |
| **KD+TLCutMix** | 75.34 (0.19) | 70.77 (0.17) | 74.91 (0.20) | 74.16 (0.18) | 68.79 (0.35) | 74.85 (0.23) | 76.61 (0.18) |
| **KD+TLCutMix+pick** | 75.59 (0.22) | 70.99 (0.20) | 74.78 (0.35) | 74.43 (0.20) | 69.49 (0.32) | 74.95 (0.18) | 76.90 (0.25) |
| **KD+TLCutMix+pick**$_{960}$ | **76.41** (0.10) | 71.66 (0.15) | **75.12** (0.18) | **75.00** (0.17) | 70.47 (0.12) | **76.13** (0.16) | **77.90** (0.30) |
| CRD (Tian et al., 2020) | 75.64 (0.21) | 71.63 (0.15) | 75.46 (0.25) | 74.29 (0.12) | 69.94 (0.05) | 74.58 (0.27) | 76.05 (0.09) |
| CRD+**TLCutMix+pick** | 75.96 (0.27) | 71.41 (0.26) | **76.11** (0.53) | 74.65 (0.12) | 69.95 (0.22) | 75.35 (0.22) | 76.93 (0.11) |
| CRD+**TLCutMix+pick**$_{960}$ | 76.61 (0.01) | 72.40 (0.20) | 75.96 (0.29) | 75.41 (0.10) | **70.84** (0.05) | 76.20 (0.22) | 78.51 (0.27) |

Table 4: Student test accuracy comparison on Tiny ImageNet. Each result is obtained by 3 random runs, mean (std) accuracy reported. The best results are in **bold** and second best underlined. The subscript 480 means the total number of training epochs (default: 480).

| Teacher
Student | WRN-40-2
WRN-16-2 | ResNet56
ResNet20 | ResNet32x4
ResNet8x4 | VGG13
VGG8 | VGG13
MobileNetV2 | ResNet50
VGG8 | ResNet32x4
ShuffleNetV2 |
|---|---|---|---|---|---|---|---|
| Teacher Acc.
Student Acc. | 61.28
58.23 | 58.37
52.53 | 64.41
55.41 | 62.59
56.67 | 62.59
58.20 | 68.20
56.67 | 64.41
62.07 |
| KD (Hinton et al., 2014) | 58.65 (0.09) | 53.58 (0.18) | 55.67 (0.09) | 61.48 (0.36) | 59.28 (0.13) | 60.39 (0.16) | 66.34 (0.11) |
| KD$_{480}$ (Hinton et al., 2014) | 59.20 (0.30) | 54.23 (0.24) | 55.49 (0.11) | 61.72 (0.10) | 59.27 (0.08) | 60.10 (0.30) | 65.81 (0.11) |
| **KD+TLCutMix** | 59.06 (0.18) | 53.77 (0.33) | 56.41 (0.04) | 62.17 (0.11) | 60.48 (0.30) | 61.12 (0.18) | 67.01 (0.30) |
| **KD+TLCutMix+pick** | 59.22 (0.05) | 53.66 (0.05) | 56.82 (0.23) | 62.32 (0.18) | 60.53 (0.18) | 61.40 (0.26) | 67.08 (0.13) |
| **KD+TLCutMix+pick**$_{480}$ | **60.07** (0.04) | **54.25** (0.07) | **57.54** (0.23) | **62.60** (0.25) | 60.66 (0.15) | **61.95** (0.14) | **67.35** (0.21) |
| CRD (Tian et al., 2020) | 60.79 (0.24) | 55.34 (0.02) | 59.28 (0.13) | 62.92 (0.31) | 62.38 (0.19) | 62.03 (0.16) | 67.33 (0.13) |
| CRD+**TLCutMix+pick** | 60.72 (0.09) | 54.99 (0.16) | 59.65 (0.24) | 63.39 (0.10) | 62.54 (0.22) | **62.85** (0.18) | 67.64 (0.18) |
| CRD+**TLCutMix+pick**$_{480}$ | **60.99** (0.33) | **55.68** (0.22) | **60.13** (0.13) | **63.60** (0.20) | **62.79** (0.03) | 62.60 (0.17) | **67.70** (0.35) |

also brings more performance gains with KD loss on WRN and VGG. The gains are more or less up to the particular pairs but the trends are consistent: When DA is used, the optimal number of training epochs is higher and even more so for KD than CE; when a stronger DA is employed, the optimal number of epochs is further significantly higher and produces lower test error. These results support the proposed hypotheses in Eq. (2).

**Exploring different data augmentation schemes**. In Tab. 1 we compare three different DA schemes on CIFAR-100: the default, TLmixup, and TLCutMix. It has been shown in the original papers of mixup and CutMix that they improve accuracy over the standard data augmentation *in the CE case*. However, it does not mean naively combining KD and CutMix/mixup as it is can always bring performance improvement. As seen, CutMix is actually at odds with KD on the ResNet56/ResNet20 pair while our TLCutMix consistently improves the performance on all three pairs. On the other pairs, original CutMix is also not as effective as our adapted TLCutMix. These confirm that using the teacher's output for distillation for the augmented data is critical. TLCutMix is better than TLmixup in general, so we choose it as base to develop TLCutMix+pick.

**Exploring different data picking schemes**. In Tab. 2, we compare the three potential schemes of selecting more informative data for the student: entropy of the teacher's output ("T ent."), entropy of the student output ("S ent."), and the KL divergence between the teacher's and student's outputs ("T/S kld"). As shown, the KL divergence scheme performs best. This is expected as either the teacher entropy or student entropy alone does not reveal the whole picture.

**Benchmark on CIFAR-100**. The results are shown in Tab. 3. We have the following observations. (1) KD can be improved by training for more iterations (960 epochs vs. 240), owing to the effect of data augmentation (only one exception is ResNet32x4/ResNet8x4). This is not true for CE alone. This is a novel observation which shows the optimal number of training epochs for KD w/ DA is significantly different from that of CE. (2) Comparing the row "KD+TLCutMix" to "KD", we see the proposed TLCutMix scheme improves the accuracies of *all* teacher-student pairs. On 5 out of

the 7 pairs, the improvement is very significant (more than 1 percentage point). (3) Comparing the row "KD+TLCutMix+pick" to "KD+TLCutMix", we see 6 out of the 7 pairs are improved further, showing the proposed data picking scheme works in most cases. (4) Finally, "KD+TLCutMix+pick" scheme can be combined with more training iterations, which delivers even higher accuracies. (5) If comparing our best results (KD+TLCutMix+pick$_{960}$) with those of CRD (though this is not an apples-to-apples comparison since the two methods focus on different aspects to improve KD), we can see our approach outperforms CRD on 6 out of the 7 pairs. It is worth emphasizing that we achieve this simply using the original KD loss (Hinton et al., 2014), *with no bells and whistles*. This justifies one of our motivations in this paper, *i.e.*, existing KD methods (Peng et al., 2019; Park et al., 2019; Tian et al., 2020) mainly improve KD at the *output* layer through better loss functions, while we propose to improve KD at the *input* end and show this path is just as promising.

In the last two rows of Fig. 3, when CRD (Tian et al., 2020), the state-of-the-art KD algorithm, is armed with our proposed "TLCutMix+pick" and more training iterations, its results can be further advanced *consistently*. This demonstrates that the proposed schemes are general and can readily work with those methods focusing on better KD loss functions. In the Appendix Tab. 9, we present the results of applying TLCutMix to another 5 KD methods. *All* of the evaluated pairs see accuracy gains; half of them are even improved by more than 1% point.

**Further remarks**. Observation (1) above has another implication to the community in addition to improving the performance of KD. It tells us that the number of training iterations can have a *big impact* on the performance of a KD method. Unaware of this issue, if authors of a KD paper compare their method to others by directly citing numbers from other papers and the training epochs happen to be different, then the comparison may well be unintentionally unfair from the beginning.

## 4.2 TINY IMAGENET

In this section we evaluate the proposed schemes on a more challenging dataset – Tiny ImageNet. Similar to the case on CIFAR-100, we have results on different teacher-student pairs, shown in Tab. 4. For more training iterations, we train for 480 epochs instead of 960 to save time. Most claims on the CIFAR-100 dataset are also validated here: (1) "KD+TLCutMix" is better than KD, which is verified on *all* pairs. (2) "KD+TLCutMix+pick" is better than "KD+TLCutMix", verified on 6 pairs. The exception pair is ResNet56/ResNet20, where adding data picking decreases the accuracy slightly by 0.11%. (3) When "KD+TLCutMix+pick" is equipped with more training iterations, we obtain the best performance. The main difference from CIFAR-100 results lies in the comparison between "KD$_{480}$" and "KD". In the CIFAR-100 case with standard augmentation, more training iterations consistently improves the accuracy on 6 pairs, while here only 3 are improved. We believe this is because the standard DA scheme – random crops and horizontal flips – cannot produce diverse enough data on this challenging dataset (Tiny ImageNet vs. CIFAR-100). In contrast, using the stronger DA scheme (TLCutMix+pick) and more training iterations *does* show significant improvement in all 7 out of 7 cases.

We also evaluate the compatibility of our DA methods with the state-of-the-art CRD, shown in the last two rows of Tab. 4. Our "TLCutMix+pick" method further advances the prior state-of-the-art on 5 pairs. When CRD+TLCutMix+pick is trained for 480 epochs (instead of 240), further improvement can be seen on 6 of 7 pairs.

## 4.3 IMAGENET

We further evaluate our methods on the ImageNet dataset, shown in Tab. 5. "KD+TLCutMix" improves original KD (from 70.66 to 71.05 in top-1 accuracy). When data picking is added, it does not help here. Possible reasons will be analyzed later. When the student is trained for 200 epochs with KD+TLCutMix+pick, it delivers *the new state-of-the-art* top-1 performance.

We also present the result of original KD trained for 200 epochs. Interestingly, it matches the previous state-of-the-art method CRD and beats many other KD methods *without any additional loss terms*. However, this is not an apples-to-apples comparison, as these methods are trained for 100 epochs. Yet it is a clear indication that the interplay between KD and DA is useful even on a large-scale dataset.

Table 5: Top-1 and Top-5 accuracy (%) of the student ResNet18 on ImageNet validation set. The subscript 200 indicates the total number of training epochs is 200 (the original one is 100).

| | Top-1 acc. | Top-5 acc. |
|---|---|---|
| Teacher (ResNet34) | 73.31 | 91.42 |
| Student (ResNet18) | 69.75 | 89.97 |
| KD (Hinton et al., 2014) | 70.66 | 89.88 |
| SP (Tung & Mori, 2019) | 70.62 | 89.80 |
| AT (Zagoruyko & Komodakis, 2017) | 70.70 | 90.00 |
| CRD (Tian et al., 2020) | 71.38 | 90.49 |
| KD$_{200}$ (Hinton et al., 2014) | 71.38 | **90.59** |
| **KD+TLCutMix** | 71.05 | 90.36 |
| **KD+TLCutMix+pick** | 70.78 | 90.04 |
| **KD+TLCutMix+pick**$_{200}$ | **71.76** | 90.58 |

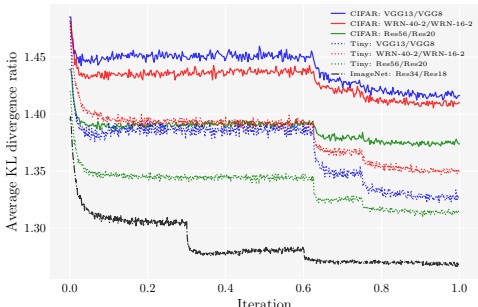

Figure 4: Mean KL divergence ratio $r$ (Eq. (4)) over iterations on different datasets. The iterations are normalized into range [0, 1] for easy comparison since the total numbers of iterations are different on the 3 datasets.

**Cross-dataset analysis**. Here we investigate how the proposed method of KD+TLCutmix+pick is affected by the size and nature of the dataset. The resnet teacher-student pairs of Res56/Res20 and Res34/Res18 are of particular interest as the boost in performance for these pairs are lower than other network architectures.

The picking scheme is proposed based on the idea of active learning (Sec. 3.2). Intuitively, it can work *only if the picked data has more information to the student network than those randomly presented*. Since we adopt the KL divergence between the teacher's output and the student's output to measure the amount of information in the input data, we can compare this metric on two different sets of data, *i.e.*, picked randomly vs. picked based on KL divergence. Specifically, we define *average KL divergence ratio*

$$r = \frac{\frac{1}{N_p} \sum_i^{N_p} d_i}{\frac{1}{N} \sum_j^{N} d_j}, \qquad (4)$$

where $d_i$ stands for the KL divergence for the $i$-th sample defined in Eq. (3); $N$ denotes the number of total samples in a batch; $N_p$ denotes the number of sample picked based on KL divergence ($N_p = N/2$ in our experiments); note that $r > 1$. Larger $r$ means the picked samples have more information than the average samples. Then we compare $r$ on different datasets over the training process of "KD+TLCutMix+pick". Results are shown in Fig. 4. As seen, in terms of $r$, CIFAR-100 > Tiny ImageNet > ImageNet on average; meanwhile, comparing the results of CIFAR-100 (Tab. 3), Tiny ImageNet (Tab. 4), and ImageNet (Tab. 5) we see the accuracy gains brought by data picking also show the *same* trend of CIFAR-100 > Tiny ImageNet > ImageNet, in accordance with our expectation. This validates the soundness of the metric $r$ we introduced. The $r$ on ImageNet is clearly lower than the other two, meaning there is no significant information difference between the picked data and the average data, which may well explain the under-performance of the data picking scheme on ImageNet. Note that the root cause of this problem actually lies in the data augmentation part – since it cannot produce more informative samples, the subsequent data-picking has no scope to expand its value. How to obtain an even stronger scheme than TLCutMix remains elusive for now, which we will investigate as part of our future work. Also note, Res56/Res20 delivers the lowest $r$ among the three pairs. This likely explains why the picking scheme is especially not effective on the original resnet pairs (Res56/Res20, Res34/Res18).

## 5 CONCLUSION

We carefully investigate the interplay between knowledge distillation (KD) and data augmentation (DA) in this paper. Unlike the cross-entropy loss, KD can exploit DA by training for more epochs. The proposed input view diversity framework explains the interplay well and inspires us to develop three new data augmentation methods specifically for KD. Extensive experiments demonstrate the merits of our methods across various networks on CIFAR-100, Tiny ImageNet, and ImageNet datasets. Our method achieves the new state-of-the-art using only the vanilla KD loss with no bells and whistles, showing the potential of improving KD from the *input* side rather than a better KD loss function. Our paper can also help the community build a more standard benchmark of KD algorithms, paying particular attention to the DA schemes and number of training epochs.

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

## A    NETWORK PARAMETERS AND FLOPS

The number of parameters and FLOPs (FLoating-point OPerations) of each model on the CIFAR-100, Tiny ImageNet, and ImageNet datasets are presented in Tab. 6, Tab. 7, and Tab. 8, respectively. For the number of parameters (or FLOPs), we only count them in the *convolutional* and *fully-connected* layers (BN layers not included) following the common practice.

**Training cost of TLCutMix+pick**. The proposed picking scheme only increases the wall-clock training time by around 30%, which is reasonable considering the performance gains.

Table 6: Model complexity statistics of each teacher-student pair on the CIFAR-100 dataset. The number of FLOPs is measured by the number of Multiply and Add operations in convolutional and fully-connected layers. Input image size: $32\times32\times3$.

| Teacher
Student | WRN-40-2
WRN-16-2 | ResNet56
ResNet20 | ResNet32x4
ResNet8x4 | VGG13
VGG8 | VGG13
MobileNetV2 | ResNet50
VGG8 | ResNet32x4
ShuffleNetV2 |
|---|---|---|---|---|---|---|---|
| Teacher #Params $(10^6)$ | 2.2497 | 0.8574 | 7.4239 | 9.4563 | 9.4563 | 23.6521 | 7.4239 |
| Student #Params $(10^6)$ | 0.7015 | 0.2768 | 1.2308 | 3.9621 | 0.7945 | 3.9621 | 1.3393 |
| Compression ratio | $3.2072\times$ | $3.0979\times$ | $6.0319\times$ | $2.3867\times$ | $11.9027\times$ | $5.9696\times$ | $5.5430\times$ |
| Teacher #FLOPs $(10^9)$ | 0.6552 | 0.2515 | 2.1661 | 0.5699 | 0.5699 | 2.5960 | 2.1661 |
| Student #FLOPs $(10^9)$ | 0.2022 | 0.0816 | 0.3541 | 0.1924 | 0.0116 | 0.1924 | 0.0863 |
| Speedup ratio | $3.2399\times$ | $3.0808\times$ | $6.1164\times$ | $2.9621\times$ | $49.1399\times$ | $13.4939\times$ | $25.0915\times$ |

Table 7: Model complexity statistics of each teacher-student pair on the Tiny ImageNet dataset. The number of FLOPs is measured by the number of Multiply and Add operations in convolutional and fully-connected layers. Input image size: $64\times64\times3$.

| Teacher
Student | WRN-40-2
WRN-16-2 | ResNet56
ResNet20 | ResNet32x4
ResNet8x4 | VGG13
VGG8 | VGG13
MobileNetV2 | ResNet50
VGG8 | ResNet32x4
ShuffleNetV2 |
|---|---|---|---|---|---|---|---|
| Teacher #Params $(10^6)$ | 2.2626 | 0.8639 | 7.4496 | 9.5076 | 9.5076 | 23.8570 | 7.4496 |
| Student #Params $(10^6)$ | 0.7144 | 0.2833 | 1.2565 | 4.0134 | 0.9226 | 4.0134 | 1.4418 |
| Compression ratio | $3.1674\times$ | $3.0498\times$ | $5.9289\times$ | $2.3690\times$ | $10.3056\times$ | $5.9444\times$ | $5.1667\times$ |
| Teacher #FLOPs $(10^9)$ | 2.6208 | 1.0060 | 8.6642 | 1.8263 | 1.8263 | 10.3833 | 8.6642 |
| Student #FLOPs $(10^9)$ | 0.8089 | 0.3265 | 1.4165 | 0.5428 | 0.0459 | 0.5428 | 0.3449 |
| Speedup ratio | $3.2400\times$ | $3.0809\times$ | $6.1168\times$ | $3.3643\times$ | $39.8097\times$ | $19.1276\times$ | $25.1210\times$ |

Table 8: Model complexity statistics of ResNet34 (teacher) / ResNet18 (student) on the ImageNet dataset. The number of FLOPs is measured by the number of Multiply and Add operations in convolutional and fully-connected layers. Input image size: $224\times224\times3$.

| Teacher #Params $(10^6)$ | Student #Params $(10^6)$ | Compression ratio | Teacher #FLOPs $(10^9)$ | Student #FLOPs $(10^9)$ | Speedup ratio |
|---|---|---|---|---|---|
| 21.7806 | 11.6799 | $1.8648\times$ | 7.3275 | 3.6281 | $2.0196\times$ |

## B    COMBINING TLCUTMIX WITH OTHER KD METHODS

In the main paper, we show that our proposed method can advance the previous state-of-the-art method (CRD) even further on most pairs (Tabs. 3 and 4). It is interesting to see if this bonus can translate to more KD methods. Therefore, here we combine TLCutMix with five more KD methods to see how it works. The five methods are AT (Zagoruyko & Komodakis, 2017), CC (Peng et al., 2019), SP (Tung & Mori, 2019), PKT (Passalis & Tefas, 2018), and VID (Ahn et al., 2019). Note, these five methods we select are the *top-performing* KD methods (besides CRD) based on the CIFAR-100 results in the CRD paper (it is easy to improve a mediocre KD method).

Results are shown in Tab. 9. When equipped with our proposed TLCutMix, *all* these methods see accuracy gains, although some teacher-student pairs see more improvement than others. For example, SP+TLCutMix and VID+TLCutMix only have marginal gains on the ResNet110/ResNet20 pair. Except them, *all the other* pairs see a *significant* accuracy improvement. There are 42 results in total in Tab. 9. **Half of them (21 pairs) are improved by more than 1% point**. Several (5 pairs) are even improved by over 2% points.

Table 9: Student test accuracy (standard deviation) of different KD methods on CIFAR-100 *when equipped with the proposed TLCutMix*. The results of the different KD methods are directly cited from the CRD paper Table 7 (Tian et al., 2020) (where they did not report the standard deviation of the accuracies except for the original KD method (Hinton et al., 2014), so the standard deviation of accuracies of these methods are missing here as well). AT, CC, SP, PKT, and VID methods include the original KD loss as part of their loss functions. Each our result is obtained by 3 random runs, mean (std) accuracy reported. Accuracy gains are colored in red (this table is *best viewed in color*).

| Teacher | WRN-40-2 | ResNet110 | ResNet32x4 | VGG13 | VGG13 | ResNet50 | ResNet32x4 |
|---|---|---|---|---|---|---|---|
| Student | WRN-16-2 | ResNet20 | ResNet8x4 | VGG8 | MobileNetV2 | VGG8 | ShuffleNetV2 |
| Teacher Acc. | 75.61 | 74.31 | 79.42 | 74.64 | 74.64 | 79.34 | 79.42 |
| Student Acc. | 73.26 | 69.06 | 72.50 | 70.36 | 64.60 | 70.36 | 71.82 |
| KD (Hinton et al., 2014) | 74.92 (0.28) | 70.67 (0.27) | 73.33 (0.25) | 72.98 (0.19) | 67.37 (0.32) | 73.81 (0.13) | 74.45 (0.27) |
| **KD+TLCutMix** (ours) | 75.34 (0.19) | 71.19 (0.23) | 74.91 (0.20) | 74.16 (0.18) | 68.79 (0.35) | 74.85 (0.23) | 76.61 (0.18) |
| Acc. gain | +0.42 | +0.52 | +1.58 | +1.18 | +1.42 | +1.04 | +2.16 |
| AT (Zagoruyko & Komodakis, 2017) | 75.32 | 70.97 | 74.53 | 73.48 | 65.13 | 74.01 | 75.39 |
| **AT+TLCutMix** (ours) | 75.65 (0.27) | 71.66 (0.07) | 75.68 (0.13) | 74.02 (0.15) | 67.20 (0.24) | 74.67 (0.14) | 76.25 (0.13) |
| Acc. gain | +0.33 | +0.69 | +1.15 | +0.54 | +2.07 | +0.66 | +0.86 |
| CC (Peng et al., 2019) | 75.09 | 70.88 | 74.21 | 73.04 | 64.86 | 73.48 | 74.71 |
| **CC+TLCutMix** (ours) | 75.75 (0.27) | 71.41 (0.24) | 75.54 (0.28) | 74.35 (0.27) | 68.44 (0.46) | 74.76 (0.09) | 76.78 (0.18) |
| Acc. gain | +0.66 | +0.53 | +1.33 | +1.31 | +0.42 | +1.28 | +2.07 |
| SP (Tung & Mori, 2019) | 74.98 | 71.02 | 74.02 | 73.49 | 68.41 | 73.52 | 74.88 |
| **SP+TLCutMix** (ours) | 75.29 (0.39) | 71.10 (0.07) | 74.96 (0.13) | 74.10 (0.27) | 68.79 (0.24) | 74.77 (0.33) | 76.24 (0.14) |
| Acc. gain | +0.31 | +0.08 | +0.94 | +0.61 | +0.38 | +1.25 | +1.36 |
| PKT (Passalis & Tefas, 2018) | 75.33 | 70.72 | 74.23 | 73.25 | 68.13 | 73.61 | 74.66 |
| **PKT+TLCutMix** (ours) | 75.85 (0.42) | 71.33 (0.06) | 75.44 (0.08) | 74.30 (0.18) | 68.98 (0.60) | 74.70 (0.32) | 76.79 (0.10) |
| Acc. gain | +0.52 | +0.61 | +1.21 | +1.05 | +0.85 | +1.09 | +2.13 |
| VID (Ahn et al., 2019) | 75.14 | 71.10 | 74.56 | 73.19 | 68.27 | 73.46 | 74.85 |
| **VID+TLCutMix** (ours) | 75.66 (0.21) | 71.13 (0.27) | 75.40 (0.17) | 74.24 (0.12) | 69.70 (0.22) | 74.67 (0.17) | 76.90 (0.17) |
| Acc. gain | +0.52 | +0.03 | +0.84 | +1.05 | +1.43 | +1.21 | +2.05 |

Table 10: KD+Cutout (DeVries & Taylor, 2017) vs. KD on the CIFAR-100 dataset.

| Teacher | WRN-40-2 | ResNet56 | ResNet32x4 | VGG13 | VGG13 | ResNet50 | ResNet32x4 |
|---|---|---|---|---|---|---|---|
| Student | WRN-16-2 | ResNet20 | ResNet8x4 | VGG8 | MobileNetV2 | VGG8 | ShuffleNetV2 |
| Teacher | 75.61 | 72.34 | 79.42 | 74.64 | 74.64 | 79.34 | 79.42 |
| Student | 73.26 | 69.06 | 72.50 | 70.36 | 64.60 | 70.36 | 71.82 |
| KD | 74.92 (0.28) | 70.66 (0.24) | 73.33 (0.25) | 72.98 (0.19) | 67.37 (0.32) | 73.81 (0.13) | 74.45 (0.27) |
| KD+Cutout | **75.54** (0.16) | **70.86** (0.19) | **74.32** (0.30) | **74.03** (0.08) | **68.22** (0.06) | **73.98 (0.19)** | **75.20** (0.10) |

As explained in the main paper, our method focuses on the *input* end to improve KD, while methods like AT, CC, SP, PKT and VID focus on the *output* end (*i.e.*, a better loss function). Therefore, they are complementary. The results here reiterate one of our contributions in this paper: most existing KD papers seek to improve KD through a better loss function while we discover a new axis – improving KD through a stronger DA, which is just as promising.

## C   KD+Cutout on CIFAR-100

In the main paper, we have shown that the KD loss can exploit more advanced data augmentation schemes (like mixup/CutMix) for improved performance. Another popular image data augmentation method, which is stronger than the common random cut and horizontal flip, is cutout (DeVries & Taylor, 2017). In Tab. 10 we show KD can also work with cutout to *consistently* deliver stronger performances. This further confirms our finding that we can simply boost the KD performance by using a stronger data augmentation.

## D   CutMix sample analysis

**CutMix sample analysis and why KD is naturally suited to exploit CutMix**. During the KD training of ResNet34/ResNet18 on ImageNet, we recorded the CutMix samples on which the teacher *disagrees* with the CutMix scheme on the label. We call this *label disagreement issue*. As show in

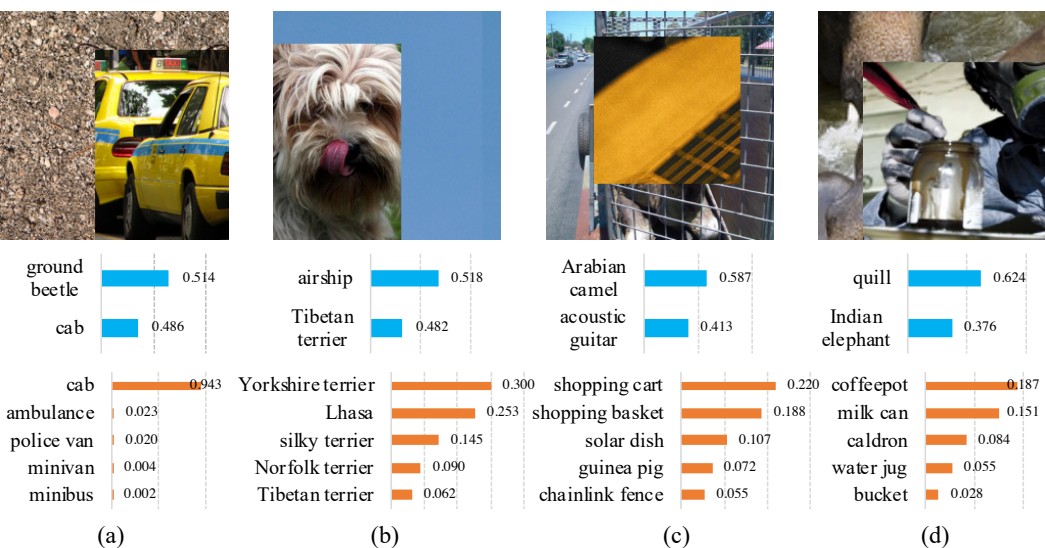

| | | | | | | | |
|---|---|---|---|---|---|---|---|
| ground beetle | 0.514 | airship | 0.518 | Arabian camel | 0.587 | quill | 0.624 |
| cab | 0.486 | Tibetan terrier | 0.482 | acoustic guitar | 0.413 | Indian elephant | 0.376 |

| | | | | | | | |
|---|---|---|---|---|---|---|---|
| cab | 0.943 | Yorkshire terrier | 0.300 | shopping cart | 0.220 | coffeepot | 0.187 |
| ambulance | 0.023 | Lhasa | 0.253 | shopping basket | 0.188 | milk can | 0.151 |
| police van | 0.020 | silky terrier | 0.145 | solar dish | 0.107 | caldron | 0.084 |
| minivan | 0.004 | Norfolk terrier | 0.090 | guinea pig | 0.072 | water jug | 0.055 |
| minibus | 0.002 | Tibetan terrier | 0.062 | chainlink fence | 0.055 | bucket | 0.028 |

| (a) | (b) | (c) | (d) |
|---|---|---|---|

Figure 5: ImageNet CutMix samples where the main object in one of the images is no longer visible after CutMix augmentation. Below each sample, the first is the target probability assigned by CutMix and the second is the top-5 predicted probabilities by the teacher. These examples can be misleading when cross-entropy loss is used, but not for KD, as explained in the text.

Fig. 5. there exist cases where the image cut from one image covers the salient object in the other. For example, the cab in (a) completely covers the ground beetle. In this case, using the label by CutMix does not make sense anymore. A similar problem appears on (b). Note that these misleading labels by CutMix are rectified when the teacher is employed to guide the student. The teacher assigns the correct label "cab" to (a) and "Yorkshire terrier" to (b) (which is still not the true label "Tibetan terrier" but it is clearly more relevant and "Tibetan terrier" is also in the top-5 predictions). For (c) and (d), they pose a problem more than occlusion: the foreground cut in (c) is labeled as "acoustic guitar", however, the cut is too small for us to make it out without knowing the label. Meanwhile, the background object "Arabian camel" is occluded. Then the grids in the picture turn out to be the most salient part. If we look at the predictions of the teacher, "shopping cart" and "shopping basket" clearly make more sense than either of the original two labels. A similar issue happens on (d), where the "Indian elephant" is largely occluded. The foreground cut is labeled "quill" but the bottle in the middle is more salient. Thus the teacher predicted it as "coffeepot", "milk can", etc. In order to see how severe the label disagreement issue is, we counted the number of these synthetic samples and found that on **more than half of the samples (52.1%)** produced by CutMix, the teacher model and CutMix hold a different view regarding the label. Many of these suffer from the problem shown in Fig. 5. The KD loss can rectify these label mistakes. This further shows the interplay between KD and DA: KD thrives on DA and *in turn, some DA schemes are more reasonable for KD* (than CE) where a teacher can supply more relevant labels.

