# OpenReview forum: "Understanding the Success of Knowledge Distillation -- A Data Augmentation Perspective"
_ICLR.cc/2022/Conference — ICLR 2022 Submitted_

### Official Review · Reviewer_146T · 2021-10-25

**Correctness:** 3
**Technical Novelty And Significance:** 3
**Empirical Novelty And Significance:** 3
**Recommendation:** 5
**Confidence:** 4

**Main Review:**

The paper has the following strengths:
S1: Explaining the rationale behind the success of KD approaches from the point of view of DA approaches is very interesting and the paper makes good progress in this aspect. Figure 2(a) is very illustrative. It clearly demonstrates that compared to applying DA approaches to CE losses, applying DA approaches to KD losses has the potential of bringing more performance gains.
S2: The experimental results presented in the paper are quite extensive and impressive. The paper conducts experiments on 3 datasets: CIFAR-100, Tiny ImageNet, and ImageNet, varying from tens of thousands of images to millions of images. The paper applies a variety of network architectures such as VGG, ResNet, Wide-ResNet, etc. to implement student models and teacher models. The paper does a wide range of ablation studies like studying effects of using more training iterations, exploring effects of using various DA approaches or data picking schemes, etc.
S3: The paper is well-presented. It is easy to follow the logic flow of the paper and understand the proposed approach.

The paper has the following weaknesses:
W1: Although the explanation in Section 3.1 why DA approaches are more beneficial to KD losses than CE losses makes sense, it is not mathematically rigorous. It is not clear how to reason from a specific example of the advantage of using DA approaches in KD losses in Figure 2(a) to the hypotheses regarding the optimal number of training epochs in Equation (2). The last two inequations in Equation (2) are sort of known hypotheses: stronger DA approaches by design can be applied to a wide range of training losses including KD losses and are expected to outperform a standard DA approach regardless of the choice of training losses. These hypotheses are disconnected from the main purpose of the paper: as indicated by the title, the paper aims to explain why KD approaches are so successful. It is highly recommended that the paper adds more theoretical details of how interplay of KD losses and CE losses is related to the success of KD.

W2: I am a bit concerned about the technical depth of the paper. Although overall the ideas make sense, these ideas are pretty straightforward. It is important for the paper to elaborate what is the challenge and how the proposed approach solves this challenge. For example, it is not clear what is the challenge of adapting stronger DA approaches when using the KD losses and the adaptation method proposed in the paper seems to be straightforward.

W3: The related work can be improved. As mentioned in the paper, an effective idea for KD is to directly match response-based knowledge, feature-based knowledge, etc. between student models and teacher models [1]. To improve the process of transferring various types of knowledge between student models and teacher models, there are many algorithms like adversarial distillation [2], multi-teacher distillation [3], etc.

W4: The proposed approach does not outperform some baseline approaches in a few experimental settings. For example, in Table 5, the baseline approach proposed by Hinton et al., 2014 performs the best on ImageNet under top-5 accuracy when using ResNet18 as student models. It would be better to explain the reason why the proposed approach under-performs the baseline approach in such experimental setting. The baseline approach by Hinton et al. is proposed around 7 years ago and it would make the conclusions more convincing if the paper could compare against more recent works [1].

[1] Knowledge Distillation: A Survey
[2] Kdgan: Knowledge distillation with generative adversarial networks
[3] Deep model compression: Distilling knowledge from noisy teachers



**Summary Of The Paper:**

This paper presents a generic framework to explain an observation that by using more training iterations, knowledge distillation (KD) losses benefit more from data augmentation (DA) than cross entropy (CE) losses do. The paper proposes an efficient DA approach to improve the performance of KD approaches by borrowing ideas from active learning. The paper shows that the proposed DA approach can improve the performance of a standard KD loss as well as more advanced KD losses. The paper also shows that the standard KD loss augmented with the proposed DA approach can outperform more advanced KD losses without using the DA approach.

The main contributions of the paper include: (1) This paper observes that DA approaches are more beneficial to KD losses than CE losses and explain the rationale behind this observation; (2) This paper proposes to enhance the performance of the tradition KD loss by adapting two stronger DA approaches named mixup and CutMix; (3) This paper further proposes an even stronger DA approach which is customized for KD losses and is built on top of ideas from active learning; (4) This paper empirically shows that incorporating the proposed DA approach into KD losses achieves new state-of-the-art accuracy on 3 datasets.


**Summary Of The Review:**

Although the paper has several strengths, I am mostly concerned about the lack of arguments supporting the main purpose of the paper and the technical depth of the paper. Also, the related work and the experimental analyses of the paper can be further improved.

---

> ### Author Response · Authors · 2021-11-22
> **Responses to Reviewer 146T (R3)**
>
> We greatly thank Reviewer 146T (R3)  for your constructive comments! Your concerns are addressed below.
>
> `R3-Q1`: why DA approaches are more beneficial to KD losses than CE losses makes sense, it is not mathematically rigorous.  It is highly recommended that the paper adds more theoretical details of how interplay of KD losses and CE losses is related to the success of KD.
>
> `R3-A1`: Admittedly, this paper is more of an empirical study than a mathematically proven theory. Thank R3 for letting us know about this problem! We shall make it more rigorous in our future version.
>
> `R3-Q2`:  I am a bit concerned about the technical depth of the paper. Although overall the ideas make sense, these ideas are pretty straightforward. It is important for the paper to elaborate what is the challenge and how the proposed approach solves this challenge. For example, it is not clear what is the challenge of adapting stronger DA approaches when using the KD losses and the adaptation method proposed in the paper seems to be straightforward.
>
> `R3-A2`: The proposed algorithm of TLCutMix is indeed not deep, while we take this simplicity as an *advantage* of our method instead of a weakness, as we show we can simply enhance the original KD loss function via a stronger DA scheme *without using any bells and whistles*. The performance improvement is non-trivial (e.g., on ImageNet, our KD+TLCutMix+pick_200 improves the original KD by over 1% top-1 accuracy). More importantly, the most valuable point of our paper, we think, is *not* one specific algorithm, but the generic framework to explain the synergistic interplay between KD and DA. That is, we find a general way that can always enhance KD performance, as long as we find a stronger DA.
>
> `R3-Q3`: The related work can be improved.
>
> `R3-A3`: Thank R3 for pointing out the related works! We'll cite these papers in our new version.
>
> `R3-Q4`: The proposed approach does not outperform some baseline approaches in a few experimental settings. For example, in Table 5, the baseline approach proposed by Hinton et al., 2014 performs the best on ImageNet under top-5 accuracy when using ResNet18 as student models. It would be better to explain the reason why the proposed approach under-performs the baseline approach in such experimental setting. The baseline approach by Hinton et al. is proposed around 7 years ago and it would make the conclusions more convincing if the paper could compare against more recent works [1].
>
> `R3-A4`: It is also pretty strange to us that our method has a better top-1 accuracy while left behind in the top-5 metric. Currectly we do not have a specific reason for this. We shall evaluate our method on more teacher-student pairs and see if we can have some insightful observations.
>
> Indeed, the original KD method is pretty old. But meanwhile, we can see this from a different perspective -- Equipped with our proposed DA scheme, the 7-year-old  baseline KD approach, now can beat CRD, which is one of the SOTA distillation methods. We believe this is a sign to show the effectiveness of our method. But anyway, greatly thank R3 for letting us know your concern. We shall include more recent KD methods in our new version to evaluate our proposed DA scheme.
>
> Hope our responses help resolve your concerns! *If you have any questions regarding our feedback, please let us know!*

---

> > ### Comment · Reviewer_146T · 2021-11-26
> > **I keep my original recommendation after reading the author response.**
> >
> > As acknowledged by the author response, this paper could be much improved by adding rigorous explanations of how interplay of KD losses and CE losses is related to the success of KD, providing insightful observations why the proposed method has a better top-1 accuracy with a worse top-5 accuracy, etc.

---

### Official Review · Reviewer_T234 · 2021-11-01

**Correctness:** 3
**Technical Novelty And Significance:** 2
**Empirical Novelty And Significance:** 3
**Recommendation:** 3
**Confidence:** 5

**Main Review:**

Strength
- A new DA technique that is effectively applied to the existing state-of-the-art knowledge distillation technique is presented.
- Many experiments prove that the proposed method is effective.
- Overall, the readability of the paper is good

Weakness
- DA is generally a widely used technique to improve the performance of DNN, and the explanation of the principle of DA's performance improvement and its effect is somewhat trivial. Therefore, contribution 1 claimed by the authors appears to be over-claimed.
- It is well known that DA differs in the degree of performance change depending on the learning configuration. However, the authors show that the proposed DA strategy is effective for only one thing: CutMix. Further analysis seems necessary.
- There are doubts as to whether the authors' DA strategy is an effective technique only for knowledge distillation, and it appears to be a general DA algorithm. For example, it is necessary to check the degree of performance improvement when the student network learns the data generated by the proposed DA strategy.
- Since the proposed DA strategy generates twice as much data as a general data provider, even if it learns the same iteration, it can be considered unfair in terms of cost.
- Comparison techniques lack the latest techniques, and it seems necessary to reinforce them through sufficient surveys.

**Summary Of The Paper:**

- This paper points out the fact that in-depth analysis of data augmentation (DA) in the field of knowledge distillation is rarely performed.
- In addition, the authors present a new DA strategy that simultaneously utilizes the data generated through the existing DA and the data generated through the stronger DA to improve the performance of the knolwedge distillation algorithm.
- And it shows that additional performance improvement is possible with a learning algorithm inspired by active learning.
- Through this, the authors show that it is possible to effectively improve performance even with the DA technique, which has not been well fused in the past.
- The proposed augmentation techniques effectively converge with existing knowledge distillation algorithms to show that they can lead to further performance improvements, and as a result, state-of-the-art performance is achieved in various datasets and various teacher-student pairs.

**Summary Of The Review:**

The analysis and proposal methods covered in this paper are interesting in terms of performance improvement.
However, it is unclear whether the DA algorithm is for knowledge distillation only.
In addition, it is necessary to analyze whether the performance improvement is due to augmentation or a large amount of learning.
Data augmentation seems to be the most important contribution point of the authors, so it seems that clear verification is needed.

---

> ### Author Response · Authors · 2021-11-22
> **Responses to Reviewer T234 (R2)**
>
> We greatly thank Reviewer T234  (R2) for your constructive comments! Your concerns are addressed below.
>
> `R2-Q1`: DA is generally a widely used technique to improve the performance of DNN, and the explanation of the principle of DA's performance improvement and its effect is somewhat trivial. Therefore, contribution 1 claimed by the authors appears to be over-claimed.
>
> `R2-A1`: This paper focuses on the interplay between KD and DA, not DA itself. Although the effectiveness of DA is well-known (actually we do not claim any contribution from the DA side alone), its interplay with KD has been less explored.
>
> `R2-Q2`: It is well known that DA differs in the degree of performance change depending on the learning configuration. However, the authors show that the proposed DA strategy is effective for only one thing: CutMix. Further analysis seems necessary.
>
> `R2-A2`: "However, the authors show that the proposed DA strategy is effective for only one thing: CutMix". With all due respect, this statement may not be true. Apart from CutMix, we also show our method is effective on mixup, the standard DA scheme (flip and crop) in the main paper, also cutout (see our appendix, Tab. 10).
>
> `R2-Q3`: There are doubts as to whether the authors' DA strategy is an effective technique only for knowledge distillation, and it appears to be a general DA algorithm. For example, it is necessary to check the degree of performance improvement when the student network learns the data generated by the proposed DA strategy.
>
> `R2-A3`: The proposed DA scheme is *not* a general DA algorithm as it demands the teacher to provide pseudo labels. This is why it is only discussed in the context of knowledge distillation instead of the general classification case. "it is necessary to check the degree of performance improvement when the student network learns the data generated by the proposed DA strategy" -- if not using the proposed DA scheme in a KD context, then the method degrades to the original CutMix (using the label assigned by the CutMix paper). Their performance has been shown in Tab. 1 (last row), where clearly, it underperforms our KD+TLCutMix, showing the merit of our proposed TLCutMix scheme.
>
> `R2-Q4`: Since the proposed DA strategy generates twice as much data as a general data provider, even if it learns the same iteration, it can be considered unfair in terms of cost.
>
> `R2-A4`: This is indeed a cost in our method. However, by our observation, this only increases the training cost by around 30%, which is still bearable given we obtain a stronger performance.
>
> `R2-Q5`: Comparison techniques lack the latest techniques, and it seems necessary to reinforce them through sufficient surveys.
>
> `R2-A5`: Despite not being the latest, CRD is one of the SOTA methods as far as we know. In this regard, our method can beat it on ImageNet *simply using the original KD loss function*, showing the strong performance of our method. Meanwhile, we shall include more SOTA KD methods to evaluate the proposed framework.
>
> Hope our responses help resolve your concerns! *If you have any questions regarding our feedback, please let us know!*

---

### Official Review · Reviewer_zS3x · 2021-11-01

**Correctness:** 2
**Technical Novelty And Significance:** 2
**Empirical Novelty And Significance:** 2
**Recommendation:** 3
**Confidence:** 4

**Main Review:**

Pros

* KD has many applications in different fields, but why it is successful is still an open research problem. The paper aims to investigate empirically and specifically the effects of DA on KD, which is a worthy problem.

* The advantage of using KD is obvious that the teacher will provide more information regarding the soft outputs, semantically similar classes that train the student better, and definitely with more examples, e.g., via data augmentation would be more useful.

* New DA method is proposed.

Cons

* The research problem is not well-stated.

* The experimental results are not sufficient to support the claims.

* The claim understanding the success of KD in DA perspective is not supported well in the paper.

* The hypothesis (Eq .2) needs more rigorous experiments to support.


**Summary Of The Paper:**

This paper aims to explore the advantage of using DA for KD for training deep neural networks and try to explain the interplay between DA and KD regarding the label advantage.

The main idea delivered in the paper regarding the interplay is that instead of fixing labels for augmented samples when using CE, the teacher in KD can act on different augmented to provide the better label to train the student. The paper also comes up with the hypothesis of the training interactions between KD and DA (e.g. KD loss makes use of more training iterations and the “stronger” DA method leads to better accuracy if training more interactions. The experiments are conducted with DA methods of Flip, Flip + Crop, TLmixup, TLCutMix. The paper also proposes TLCutMix+pick, which improves the baseline TLCutMix with an active learning approach.

Experiments conducted on CIFAR-100, Tiny ImageNet and ImageNet show the improvements of using DA over the baselines and the improvement of the proposed TLCutMix+pick.

**Summary Of The Review:**

The paper investigate the interesting problem, but more rigorous experiments and clear statement stated in the paper. The improvements of the paper are not substantial and few of definition and hypothesis are vague and not well-supported.

Details of Cons

* The key concern about the paper is the question being tackled in the paper is unclear, and the lack of rigorous experimentation to support the hypothesis.  For example, the purpose of investigating interplay is not well-stated. Why is it important and useful? Also, after reading the paper I do not understand the conclusion after investigating the interplay between KD and DA?

* The main things shown empirically in the paper are that DA is helpful to improve accuracy, which is different from the claim of the paper in understanding the success of KD via the DA perspective. rather than that, no other insights about understanding more about KD or fundamental reasons why DA improves KD are not clear (at least no empirical support). We observe data augmentation can improve most training models, it is reasonable and not too surprising it will improve the student model. The question is that the student model is improved because of simply more training data to be used or because the soft label provided by the teacher makes the training better (we observe in the baseline CE, simply adding more data performance gets improved)? One simple experiment to demonstrate that is to modify the current model using the soft-label of the original samples as the soft-label of the augmented sample in KD training.

* Another concern about the performance is that the improvements are minor compared to the original KD and most of the experiments are less than 1%.

* There are also concerns about the hypothesis (Eq. 2). I believe it would need rigorous experiments to demonstrate this because it depends a lot on the network architecture and dataset size as well as the data augmentation methods and hyper-parameters. Without that, it is difficult to make the final conclusion. Moreover, I wonder why and how this hypothesis is useful for the community to investigate? I understand when we have more training data provided to the model it will take more time for the model to  converge. By the way, not sure how the authors defined a “stronger” DA? It looks like the “stronger” one will give better accuracy to students which may be vague.

* As claimed in the paper, the “stronger” DA will need to train more iterations to achieve good accuracy. Does the author consider the combination of multiple DAs together a “strong” DA?

* In the framework, can the author explain the reason to keep x and x’? Does that help to boost the performance? As in Eq. 1 we already have the CE part using the original samples x?

* Many related state of the art data augmentations are missing:

[1] AutoAugment: Learning Augmentation Strategies from Data.
[2] RandAugment
[3] Fast AutoAugment.  ​

Minor comments:

Figures and plots are not high quality.

Eq. 1 is the combination of CE and KD; the hyper parameter $\alpha$ may lead to different observations and behaviors. Would be good to study on that.

---

> ### Author Response · Authors · 2021-11-22
> **Responses to Reviewer zS3x (R1)**
>
> We greatly thank Reviewer zS3x (R1) for your constructive comments! Your concerns are addressed below.
>
> `R1-Q1`: The key concern about the paper is the question being tackled in the paper is unclear, and the lack of rigorous experimentation to support the hypothesis. For example, the purpose of investigating interplay is not well-stated. Why is it important and useful? Also, after reading the paper I do not understand the conclusion after investigating the interplay between KD and DA?
>
> `R1-A1`: When we say "the success of knowledge distillation". Implicitly, we mean "the success of knowledge distillation" against the CE (cross-entropy) loss. Then, previous works mainly think this is attributed to the "dark knowledge" hidden in the output classes. Namely, they analyze it from the network *output* side, while in this paper, we analyze it from the *input* side, which states, KD loss can harvest more gains than CE loss from the *same* data augmentation. That is, the success of KD is also partly attributed to the fact that *the KD loss is more effective in exploiting the data provided by a certain DA scheme*. To our best knowledge, this has not been proposed before. It is important because (1) we can better understand why KD loss performs better than CE loss (2) it can help us achieve stronger KD performance even not changing the loss function (which is also the reason it is *useful*).
>
> `R1-Q2`: why DA improves KD are not clear.  We observe data augmentation can improve most training models, it is reasonable and not too surprising it will improve the student model.
>
> `R1-A2`: The statements of "DA improves KD" or "DA improves the student model" are talking about the effectiveness of DA. They are well-known, admittedly (see page 2, "The first two observations are well-recognized by existing works (they simply reiterate the effectiveness of KD and DA, respectively"). This paper focuses on the *relative* advantage of KD over CE under the same DA. Especially, with a *stronger* DA, this relative advantage is *more* pronounced.
>
> `R1-Q3`: Another concern about the performance is that the improvements are minor compared to the original KD and most of the experiments are less than 1%.
>
> `R1-A3`: On CIFAR-100, our method is better than the KD counterpart (comparing "KD+TLCutMix+pick_960" vs "KD_960" or "KD+TLCutMix+pick" vs. "KD") by over 1% on *5/7 pairs*: ResNet32x4/ResNet8x4, VGG13/VGG8, VGG13/MobileV2, Res50/VGG8, ResNet32x4/ShuffleNetV2. R1's concern seems to rest on our ImageNet experiment, where KD is well-known not so effective itself. Yet our method still improves KD under this hard case by around 0.4% top-1 accuracy, which is a non-trivial improvement on ImageNet. Also, the ImageNet result is the *new SOTA* KD result simply using the original KD loss, to our best knowledge.
>
> `R1-Q4`: By the way, not sure how the authors defined a “stronger” DA? It looks like the “stronger” one will give better accuracy to students which may be vague.
>
> `R1-A4`: This is indeed an open question in our paper. Currently, it is defined by a stronger classification accuracy. We'll explore this further in our future work.
>
> `R1-Q5`: Does the author consider the combination of multiple DAs together a “strong” DA?
>
> `R1-A5`: Yes! This is definitely in our consideration. E.g., for the Fig.1, we show the combination of Flip+Crop, which is stronger than Flip alone, and delivers better KD performance as expected. How to combine even more DA schemes effectively falls into our future work considering there are so many DA schemes currently.
>
>  `R1-Q6`: In the framework, can the author explain the reason to keep x and x’? Does that help to boost the performance? As in Eq. 1 we already have the CE part using the original samples x?
>
> `R1-A6`: We keep x for the reason explained on page 5: "The consideration of keeping both inputs is to maintain the information path for the original input x so that we can easily see how the added information path of x' leads to a difference." Although CE loss already includes x, yet it is for *CE* itself; x can be further exploited in the *KL divergence* loss term, thus intuitively keeping x is also beneficial. In practice, we do see keeping x helps boost the performance.
>
> `R1-Q7`: Many related state of the art data augmentations are missing.
>
> `R1-A7`: Thank R1 for pointing out these related works. We shall cite them in our new version.
>
> `R1-Q8`: the hyper parameter $\alpha$ may lead to different observations and behaviors. Would be good to study on that.
>
> `R1-A8`: The alpha is inherited from the CRD paper, which is a pretty standard setting for KD. We will include study results on it in our new version.
>
> Hope our responses help resolve your concerns! *If you have any questions regarding our feedback, please let us know!*

---

### Decision · Program_Chairs · 2022-01-20

**Decision:**

Reject

**Comment:**

All of the reviewers recommended rejecting this paper.
There were concerns that the underlying research questions being probed were not expressed clearly enough.
Reviewers were concerned that the experimental work was not sufficient to warrant acceptance.
Other concerns included the technical depth of the paper, the degree to which related work was discussed, placed in context and compared with empirically.
The AC recommends rejecting this paper.